# SOSELETO: A Unified Approach to Transfer Learning and Training with Noisy Labels

**Or Litany**[*]
Facebook AI Research
Menlo Park, CA, USA
orlitany@gmail.com

**Daniel Freedman**
Google Research
Haifa, Israel
danielfreedman@google.com

## Abstract

We present SOSELETO (SOurce SELEction for Target Optimization), a new method for exploiting a source dataset to solve a classification problem on a target dataset. SOSELETO is based on the following simple intuition: some source examples are more informative than others for the target problem. To capture this intuition, source samples are each given weights; these weights are solved for jointly with the source and target classification problems via a bilevel optimization scheme. The target therefore gets to choose the source samples which are most informative for its own classification task. Furthermore, the bilevel nature of the optimization acts as a kind of regularization on the target, mitigating overfitting. SOSELETO may be applied to both classic transfer learning, as well as the problem of training on datasets with noisy labels; we show state of the art results on both of these problems. Code has been made available at: https://github.com/orlitany/SOSELETO.

## 1 Introduction

Deep learning has demonstrated remarkable successes in tasks where large training sets are available. Yet, its usefulness is still limited in many important problems that lack such data. A natural question is then how one may apply the techniques of deep learning within these relatively data-poor regimes. A standard approach that seems to work relatively well is *transfer learning*. Despite its success, we claim that this approach misses an essential insight: *some source examples are more informative than others for the target classification problem.* Unfortunately, we don't know *a priori* which source examples will be important. Thus, we propose to learn this source filtering as part of an end-to-end training process.

The resulting algorithm is SOSELETO: SOurce SELEction for Target Optimization. Each training sample in the source dataset is given a weight, representing its importance. A shared source/target representation is then optimized by means of a bilevel optimization. In the interior level, the source minimizes its classification loss with respect to the representation and classification layer parameters, for fixed values of the sample weights. In the exterior level, the target minimizes its classification loss with respect to both the source sample weights and its own classification layer. The sample weights implicitly control the representation through the interior level. The target therefore gets to choose the source samples which are most informative for its own classification task. Furthermore, the bilevel nature of the optimization acts as a kind of regularization on the target, mitigating overfitting, as the target does not directly control the representation parameters. The entire process – training of the shared representation, source and target classifiers, and source weights – happens simultaneously.

**Related Work** The most common techniques for transfer learning are feature extraction e.g. Donahue et al. (2014) and fine-tuning, e.g. Girshick et al. (2014). An older survey of transfer learning techniques may be found in Pan & Yang (2010). Domain adaptation (Saenko et al., 2010) involves knowledge transfer when the source and target classes are the same. Earlier techniques aligned the source and target via matching of feature space statistics (Tzeng et al., 2014; Long et al., 2015); subsequent work used adversarial methods to improve the domain adaptation performance (Ganin & Lempitsky, 2015; Tzeng et al., 2015; 2017; Hoffman et al., 2017). In this paper, we are more interested in transfer learning where the source and target classes are different. Long et al. (2017); Pei et al. (2018); Cao et al. (2018a;b) address domain adaptation that is closer to our setting. Cao

---

[*]The majority of the work was done while the author was at Google Research.

et al. (2018b) examines "partial transfer learning" in which there is partial overlap between source and target classes (often the target classes are a subset of the source). This setting is also dealt with in Busto & Gall (2017). Like SOSELETO, Ge & Yu (2017) propose selecting a portion of the source dataset, however, the selection is done prior to training and is not end-to-end. In Luo et al. (2017), an adversarial loss aligns the source and target representations in a few-shot setting.

Instance reweighting is a well studied technique for domain adaptation, demonstrated e.g. in Covariate Shift methods (Shimodaira, 2000; Sugiyama et al., 2007; 2008). While in these works, the source and target label spaces are the same, we allow them to be different – even entirely non-overlapping. Crucially, we do not make assumptions on the similarity of the *distributions* nor do we explicitly optimize for it. The same distinction applies for the recent work of Yan et al. (2017), and for the partial overlap assumption of Zhang et al. (2018). In addition, these two works propose an *unsupervised* approach, whereas our proposed method is completely supervised.

Classification with noisy labels is a longstanding problem in the machine learning literature, see the review paper Frénay & Verleysen (2014). Within the realm of deep learning, it has been observed that with sufficiently large data, learning with label noise – without modification to the learning algorithms – actually leads to reasonably high accuracy (Krause et al., 2016; Sun et al., 2017; Rolnick et al., 2017; Drory et al., 2018). We consider the setting where the large noisy dataset is accompanied by a small clean dataset. Sukhbaatar et al. (2014) introduced a noise layer into the CNN that adapts the output to align with the noisy label distribution. Xiao et al. (2015) proposed to predict simultaneously the clean label and the type of noise; Li et al. (2017) consider the same setting, but with additional information in the form of a knowledge graph on labels. Malach & Shalev-Shwartz (2017) conditioned the gradient propagation on the agreement of two separate networks. Liu & Tao (2016); Yu et al. (2017) combine ideas of learning with label noise with instance reweighting.

## 2  SOSELETO: SOurce SELEction for Target Optimization

We consider the problem of classifying a data-poor target set, by utilizing a data-rich source set. The sets and their corresponding labels are denoted by $\{(x_i^s, y_i^s)\}_{i=1}^{n^s}$, and $\{(x_i^t, y_i^t)\}_{i=1}^{n^t}$ respectively, where, $n^t \ll n^s$. The key insight is that not all source examples contribute equally useful information in regards to the target problem. For example, suppose that the source set consists of a broad collection of natural images; whereas the target set consists exclusively of various breeds of dog. We would assume that images of dogs, as well as wolves, cats and even objects with similar textures like rugs in the source set may help in the target classification task; On the flip side, it is probably less likely that images of airplanes and beaches will be relevant (though not impossible). However, the idea is not to come with any preconceived notions (semantic or otherwise) as to which source images will help; rather, the goal is to let the algorithm choose the relevant source images, in an end-to-end fashion.

We assume that the source and target networks share the same architecture, except for the last classification layers. In particular, the architecture is given by $F(x; \theta, \phi)$, where $\phi$ denotes the last classification layer(s), and $\theta$ constitutes all of the "representation" layers. The source and target networks are thus $F(x; \theta, \phi^s)$, and $F(x; \theta, \phi^t)$, respectively. By assigning a weight $\alpha_j \in [0, 1]$ to each source example, we define the *weighted* source loss as: $L_s(\theta, \phi^s, \alpha) = \frac{1}{n^s} \sum_{j=1}^{n^s} \alpha_j \ell(y_j^s, F(x_j^s; \theta, \phi^s))$ where $\ell(\cdot, \cdot)$ is a per example classification loss (e.g. cross-entropy). The weights $\alpha_j$ will allow us to decide which source images are most relevant for the target classification task. The target loss is standard: $L_t(\theta, \phi^t) = \frac{1}{n^t} \sum_{i=1}^{n^t} \ell(y_i^t, F(x_i^t; \theta, \phi^t))$. As noted in Section 1, this formulation allows us to address both the transfer learning problem as well as learning with label noise. In the former case, the source and target may have non-overlapping label spaces; high weights will indicate which source examples have relevant knowledge for the target classification task. In the latter case, the source is the noisy dataset, the target is the clean dataset, and they share a classifier (i.e. $\phi^t = \phi^s$) as well as a label space; high weights will indicate which source examples are likely to be reliable. In either case, the target is much smaller than the source.

The question now becomes: how can we combine the source and target losses into a single optimization problem? A simple idea is to create a weighted sum of source and target losses. Unfortunately, issues are likely to arise regardless of the weight chosen. If the target is weighted equally to the source, then overfitting may likely result given the small size of the target. On the other hand, if the weights are proportional to the size of the two sets, then the source will simply drown out the target.

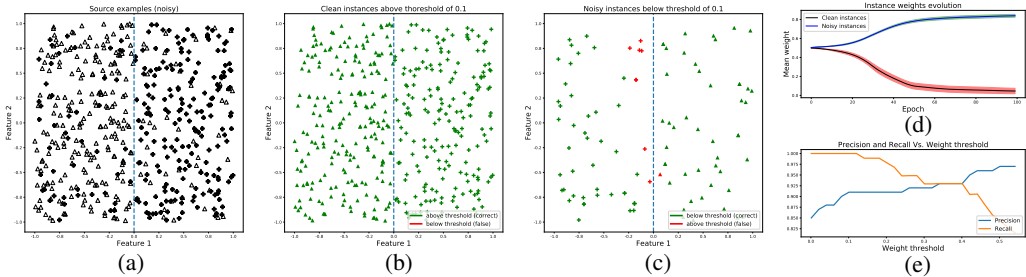

Figure 1: SOSELETO applied to a synthetic noisy labels problem – see text for details.

We propose instead to use *bilevel optimization*. Specifically, in the interior level we find the optimal features and source classifier as a function of the weights $\alpha$, by minimizing the source loss:

$$\theta^*(\alpha), \phi^{s*}(\alpha) = \arg\min_{\theta, \phi^s} L_s(\theta, \phi^s, \alpha) \tag{1}$$

In the exterior level, we minimize the target loss, but only through access to the source weights; that is, we solve:

$$\min_{\alpha, \phi^t} L_t(\theta^*(\alpha), \phi^t) \tag{2}$$

Why might we expect this bilevel formulation to succeed? The key is that the target only has access to the features in an *indirect* manner, by controlling which source examples will dominate the source classification problem. This form of regularization, mitigates overfitting, which is the main threat when dealing with a small set such as the target.

We adopt a stochastic approach for optimizing the bilevel problem. Specifically, at each iteration, we take a gradient step in the interior level problem (1):

$$\theta_{m+1} = \theta_m - \lambda_p \frac{\partial L_s}{\partial \theta}(\theta_m, \phi_m^s, \alpha_m) = \theta_m - \lambda_p Q(\theta_m, \phi_m^s)\alpha_m \tag{3}$$

and $Q(\theta, \phi^s)$ is a matrix whose $j^{th}$ column is given by $q_j = \frac{1}{n^s}\frac{\partial}{\partial \theta}\ell(y_j^s, F(x_j^s; \theta, \phi^s))$. An identical descent equation exists for the classifier $\phi^s$, which we omit for clarity.

Given this iterative version of the interior level of the bilevel optimization, we may now turn to the exterior level. Plugging Equation (3) into Equation (2) gives: $\min_{\alpha, \phi^t} L_t(\theta_m - \lambda_p Q\alpha, \phi^t)$. We can then take a gradient descent step of this equation, yielding:

$$\alpha_{m+1} = \alpha_m - \lambda_\alpha \frac{\partial}{\partial \alpha} L_t(\theta_m - \lambda_p Q\alpha, \phi^t) \approx \alpha_m + \lambda_\alpha \lambda_p Q^T \frac{\partial L_t}{\partial \theta}(\theta_m) \tag{4}$$

where we have made use of the fact that $\lambda_p$ is small. Of course, there will also be a descent equation for the classifier $\phi^t$. The resulting update scheme is quite intuitive: source example weights are update according to how well they align with the target aggregated gradient. Finally, we clip the weight values to be in the range $[0, 1]$ (a detailed explanation to this step is provided in Appendix B). For completeness we have included a summary of SOSELETO algorithm in Appendix A. Convergence properties are discussed in Appendix C.

## 3 RESULTS

**Noisy Labels: Synthetic Experiment** We illustrate how SOSELETO works on the problem of learning with noisy labels, on a synthetic example, see Figure 1. (a) The source set consists of 500 points in $\mathbb{R}^2$, separated into two classes by the $y$-axis. However, 20% of the points are are assigned a wrong label. The target set consists of 50 "clean" points (For clarity, the target set is not illustrated.) SOSELETO is run for 100 epochs. In Figures 1(b) and 1(c), we choose a threshold of 0.1 on the weights $\alpha$, and colour the points accordingly. In particular, in Figure 1(b) the clean (i.e. correctly labelled) instances which are above the threshold are labelled in green, while those below the threshold are labelled in red; as can be seen, all of the clean points lie above the threshold for this choice of threshold, meaning that SOSELETO has correctly identified all of the clean points. In Figure 1(c), the noisy (i.e. incorrectly labelled) instances which are *below* the threshold are labelled in green; and those above the threshold are labelled in red. In this case, SOSELETO correctly identifies most of these noisy labels by assigning them small weights, while the few mislabeled

| Noise Level | CIFAR-10 Quick − | Sukhbaatar et al. (2014) 10K clean examples | Xiao et al. (2015) 10K clean examples | Ours 5K clean examples |
|---|---|---|---|---|
| 30% | 65.57 | 69.73 | 69.81 | **72.41** |
| 40% | 62.38 | 66.66 | 66.76 | **69.98** |
| 50% | 57.36 | 63.39 | 63.00 | **66.33** |

Table 1: Noisy labels: CIFAR-10. Best results in bold.

points (shown in red), are all near the separator. In Figure 1(d), a plot is shown of mean weight vs. training epoch for clean instances and noisy instances. As expected, the algorithm nicely separates the two sets. Figure 1(e) demonstrates that the good behaviour is fairly robust to the threshold chosen.

**Noisy Labels: CIFAR-10**  We now turn to a real-world setting of the problem of learning with label noise using a noisy version of CIFAR-10 (Krizhevsky & Hinton, 2009), following the settings used in Sukhbaatar et al. (2014); Xiao et al. (2015). The train set consists of $50K$ images, of which both Sukhbaatar et al. (2014); Xiao et al. (2015) set aside $10K$ clean examples for pre-training, a necessary step in both of these algorithms. In contrast, we use half that size. We also compare with a baseline of training only on noisy labels. In all cases, "Quick" cif architecture has been used. We set: $\lambda_p = 10^{-4}$, target and source batch-sizes: 32, and 256. Performance on a $10K$ test set are shown in Table 1 for three noise levels. Our method surpasses previous methods on all three noise levels. We provide further analysis in Appendix D.

**Transfer Learning: SVHN 0-4 to MNIST 5-9**  We now examine the performance of SOSELETO on a transfer learning task, using a challenging setting where: (a) source and target sets have disjoint label sets, and (b) the target set is very small. The source dataset is chosen as digits 0-4 of SVHN (Netzer et al., 2011). The target dataset is a very small subset of digits 5-9 from MNIST LeCun et al. (1998): either 20 or 25 images. Having no overlap between source and target classes renders this task transfer learning (rather than domain adaptation).

We compare our results with the following techniques: target only, which indicates training on just the target set; standard fine-tuning; Matching Nets Vinyals et al. (2016) and a fine-tuned version thereof; and two variants of the Label Efficient Learning technique described in Luo et al. (2017). Except for the target only and fine-tuning approaches, all other approaches *rely on a huge number of unlabelled target data*. By contrast, we do not make use of any of this data.

For each of the above methods, the simple LeNet architecture LeCun et al. (1998) was used. We set: $\lambda_p = 10^{-2}$, source and the target batch-sizes: 32, 10. The performance of the various methods on MNIST's test set is shown in Table 2, split into two parts: techniques which use the 30K examples of unlabelled data, and those which do not. SOSELETO has superior performance to all techniques except the Label Efficient. In Appendix E we further analyze which SVHN instances are considered more useful than others by SOSELETO in a partial class overlap setting, namely transferring SVHN 0-9 to MNIST 0-4.

Although not initially designed to use unlabelled data, one may do so using the learned SOSELETO classifier to classify the unlabelled data, generating noisy labels. SOSELETO can now be run in a label-noise mode. In the case of $n^t = 25$, this procedure elevates the accuracy to above **92**%.

| Uses Unlabelled Data? | Method | $n^t = 20$ | $n^t = 25$ |
|---|---|---|---|
| No | Target only | 80.1 | 84.0 |
| No | Fine-tuning | 80.2 | 83.0 |
| No | SOSELETO | **83.2** | **87.9** |
| Yes | Vinyals et al. (2016) | 56.6 | 51.3 |
| Yes | Fine-tuned variant of Vinyals et al. (2016) | 79.3 | 82.7 |
| Yes | Luo et al. (2017) | 80.4 | 83.1 |
| Yes | Label-efficient version of Luo et al. (2017) | **94.2** | **95.0** |

Table 2: SVHN 0-4 → MNIST 5-9. Best results in bold.

## 4    CONCLUSIONS

We have presented SOSELETO, a technique for exploiting a source dataset to learn a target classification task. This exploitation takes the form of joint training through bilevel optimization, in which the source loss is weighted by sample, and is optimized with respect to the network parameters; while the target loss is optimized with respect to these weights and its own classifier. We have empirically shown the effectiveness of the algorithm on both learning with label noise, as well as transfer learning problems. An interesting direction for future research involves incorporating an additional domain alignment term. We note that SOSELETO is architecture-agnostic, and may be extended beyond classification tasks.

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

## APPENDIX A    ALGORITHM SUMMARY

---

**Algorithm 1** SOSELETO: SOurce SELEction for Target Optimization

---

Initialize: $\theta, \phi^s, \alpha, \phi^t$.
**while** not converged **do**
    Sample source batch $b \leftarrow \{b_1, \ldots, b_L\} \subset \{1, \ldots, n^s\}$
    Denote by $\alpha_b = [\alpha_{b_1}, \ldots, \alpha_{b_L}]$
    $Q \leftarrow [q_1 \ldots q_L]$   where   $q_\ell \leftarrow \frac{1}{n^s} \frac{\partial}{\partial \theta} \ell(y^s_{b_\ell}, F(x^s_{b_\ell}; \theta, \phi^s))$
    $R \leftarrow [r_1 \ldots r_L]$   where   $r_\ell \leftarrow \frac{1}{n^s} \frac{\partial}{\partial \phi^s} \ell(y^s_{b_\ell}, F(x^s_{b_\ell}; \theta, \phi^s))$
    $\theta \leftarrow \theta - \lambda_p Q \alpha_b$
    $\phi^s \leftarrow \phi^s - \lambda_p R \alpha_b$
    $\alpha_b \leftarrow \text{CLIP}_{[0,1]} \left( \alpha_b + \lambda_\alpha \lambda_p Q^T \frac{\partial L_t}{\partial \theta} \right)$
    $\phi^t \leftarrow \phi^t - \lambda_p \frac{\partial L_t}{\partial \phi^t}$
**end while**

---

SOSELETO consists of alternating the interior and exterior descent operations, along with the descent equations for the source and target classifiers $\phi^s$ and $\phi^t$. As usual, the whole operation is done on a mini-batch basis, rather than using the entire set; note that if processing is done in parallel, then source mini-batches are taken to be non-overlapping, so as to avoid conflicts in the weight updates. A summary of SOSELETO algorithm appears in 1. Note that the target derivatives $\partial L_t / \partial \theta$ and $\partial L_t / \partial \phi^t$ are evaluated over a target mini-batch; we suppress this for clarity.

In terms of time-complexity, we note that each iteration requires both a source batch and a target batch; assuming identical batch sizes, this means that SOSELETO requires about twice the time as the ordinary source classification problem. Regarding space-complexity, in addition to the ordinary network parameters we need to store the source weights $\alpha$. Thus, the additional relative space-complexity required is the ratio of the source dataset size to the number of network parameters. This is obviously problem and architecture dependent; a typical number might be given by taking the source dataset to be Imagenet ILSVRC-2012 (size 1.2M) and the architecture to be ResNeXt-101 Xie et al. (2017) (size 44.3M parameters), yielding a relative space increase of about 3%.

## APPENDIX B    CONSTRAINING THE WEIGHTS

Recall that our goal is to explicitly require that $\alpha_j \in [0, 1]$. We may achieve this by requiring

$$\alpha_j = \sigma(\beta_j) = \begin{cases} 0 & \text{if } \beta_j < 0 \\ \beta_j & \text{if } 0 \le \beta_j \le 1 \\ 1 & \text{if } \beta_j > 1 \end{cases}$$

where the new variable $\beta_j \in \mathbb{R}$, and $\sigma(\cdot)$ is a kind of piecewise linear sigmoid function.

Now we will wish to replace the Update Equation (4), the update for $\alpha$, with a corresponding update equation for $\beta$. This is straightforward. Define the Jacobian $\partial \alpha / \partial \beta$ by

$$\left( \frac{\partial \alpha}{\partial \beta} \right)_{ij} = \frac{\partial \alpha_i}{\partial \beta_j}$$

Then we modify Equation (4) to read

$$\beta_{m+1} = \beta_m + \lambda_\alpha \lambda_p \left( \frac{\partial \alpha}{\partial \beta} \right)^T Q^T \frac{\partial L_t}{\partial \theta}(\theta_m)$$

The Jacobian is easy to compute analytically:

$$\frac{\partial \alpha}{\partial \beta} = \text{diag}(\sigma'(\beta_j)), \quad \text{where} \quad \sigma'(z) = \begin{cases} 0 & \text{if } z < 0 \\ 1 & \text{if } 0 \le z \le 1 \\ 0 & \text{if } z > 1 \end{cases}$$

Due to this very simple form, it is easy to see that $\beta_m$ will never lie outside of $[0, 1]$; and thus that $\alpha_m = \beta_m$ for all time. Hence, we can simply replace this equation with

$$\alpha_{m+1} = \text{CLIP}_{[0,1]} \left( \alpha_m + \lambda_\alpha \lambda_p Q^T \frac{\partial L_t}{\partial \theta}(\theta_m) \right)$$

where $\text{CLIP}_{[0,1]}$ clips the values below 0 to be 0; and above 1 to be 1.

## APPENDIX C   PROOF OF CONVERGENCE

SOWETO is only an approximation to the solution of a bilevel optimization problem. As a result, it is not entirely clear whether it will even converge. In this section, we demonstrate a set of sufficient conditions for SOWETO to converge to a local minimum of the target loss $L_t$.

To this end, let us examine the change in the target loss from iteration $m$ to $m + 1$:

$$\begin{aligned}
\Delta L_t &= L_t(\theta_{m+1}, \phi^t_{m+1}) - L_t(\theta_m, \phi^t_m) \\
&= L_t \left( \theta_m - \lambda_p Q \alpha_m \, , \, \phi^t_m - \lambda_p \frac{\partial L_t}{\partial \phi^t} \right) - L_t(\theta_m, \phi^t_m) \\
&\approx L_t(\theta_m, \phi^t_m) - \lambda_p \left( \frac{\partial L_t}{\partial \theta} \right)^T Q \alpha_m - \lambda_p \left( \frac{\partial L_t}{\partial \phi^t} \right)^T \frac{\partial L_t}{\partial \phi^t} - L_t(\theta_m, \phi^t_m) \\
&= -\lambda_p \left( \frac{\partial L_t}{\partial \theta} \right)^T Q \alpha_m - \lambda_p \left\| \frac{\partial L_t}{\partial \phi^t} \right\|^2
\end{aligned}$$

Now, we can use the evolution of the weights $\alpha$. Specifically, we substitute Equation (4) into the above, to get

$$\begin{aligned}
\Delta L_t &\approx -\lambda_p \left( \frac{\partial L_t}{\partial \theta} \right)^T Q \left( \alpha_{m-1} + \lambda_\alpha \lambda_p Q^T \frac{\partial L_t}{\partial \theta} \right) - \lambda_p \left\| \frac{\partial L_t}{\partial \phi^t} \right\|^2 \\
&= -\lambda_p \left( \frac{\partial L_t}{\partial \theta} \right)^T Q \alpha_{m-1} - \lambda_\alpha \lambda_p^2 \left\| Q^T \frac{\partial L_t}{\partial \theta} \right\|^2 - \lambda_p \left\| \frac{\partial L_t}{\partial \phi^t} \right\|^2 \\
&\equiv \Delta L_t^{FO}
\end{aligned}$$

where $\Delta L_t^{FO}$ indicates the change in the target loss, to first order.

Note that the latter two terms in $\Delta L_t^{FO}$ are both negative, and will therefore cause the first order approximation of the target loss to decrease, as desired. As regards the first term, matters are unclear. However, it is clear that if we set the learning rate $\lambda_\alpha$ sufficiently large, the second term will eventually dominate the first term, and the target loss will be decreased. Indeed, we can do a slightly finer analysis. Ignoring the final term (which is always negative), and setting $v = Q^T \frac{\partial L_t}{\partial \theta}$, we have that

$$\begin{aligned}
\Delta L_t^{FO} &\leq -\lambda_p v^T \alpha_{m-1} - \lambda_\alpha \lambda_p^2 \|v\|^2 \\
&\leq \lambda_p \|v\|_1 - \lambda_\alpha \lambda_p^2 \|v\|_2^2 \\
&\leq \lambda_p \sqrt{n^s} \|v\|_2 - \lambda_\alpha \lambda_p^2 \|v\|_2^2 \\
&= \lambda_p \|v\|_2 \left( \sqrt{n^s} - \lambda_\alpha \lambda_p \|v\|_2 \right)
\end{aligned}$$

where in the second line we have used the fact that all elements of $\alpha$ are in $[0, 1]$; and in the third line, we have used a standard bound on the $L_1$ norm of a vector.

Thus, a sufficient condition for the first order approximation of the target loss to decrease is if

$$\lambda_\alpha \geq \frac{\sqrt{n^s}}{\lambda_p \left\| Q^T \frac{\partial L_t}{\partial \theta} \right\|}$$

If this is true at all iterations, then the target loss will continually decrease and converge to a local minimum (given that the loss is bounded from below by 0).

## APPENDIX D   CIFAR-10 WITH NOISY LABELS – FURTHER ANALYSIS

We perform further analysis by examining the $\alpha$-values that SOSELETO chooses on convergence, see Figure 2. To visualize the results, we imagine thresholding the training samples in the source set on the basis of their $\alpha$-values; we only keep those samples with $\alpha$ greater than a given threshold. By increasing the threshold, we both reduce the total number of samples available, as well as change the effective noise level, which is the fraction of remaining samples which have incorrect labels. We may therefore plot these two quantities against each other, as shown in Figure 2; we show three plots, one for each noise level. Looking at these plots, we see for example that for the 30% noise level, if we take the half of the training samples with the highest $\alpha$-values, we are left with only about 4% which have incorrect labels. We can therefore see that SOSELETO has effectively filtered out the incorrect labels in this instance. For the 40% and 50% noise levels, the corresponding numbers are about 10% and 20% incorrect labels; while not as effective in the 30% noise level, SOSELETO is still operating as designed. Further evidence for this is provided by the large slopes of all three curves on the righthand side of the graph.

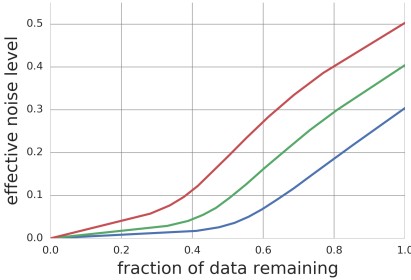

Figure 2: Noisy labels on CIFAR-10: Effect of $\alpha$-values chosen by SOSELETO. Blue is 30% noise, green is 40% noise, red is 50% noise. See accompanying explanation in the text.

## APPENDIX E   ANALYZING SVHN 0-9 TO MNIST 5-9

SOSELETO is capable of automatically pruning unhelpful instances at train time. The experiment presented in Section 3 demonstrates how SOSELETO can improve classification of MNIST 5-9 by making use of different digits from a different dataset (SVHN 0-4). To further reason about which instances are chosen as useful, we have conducted another experiment: SVHN 0-9 to MNIST 5-9. There is now a partial overlap in classes between source and target. Our findings are summarized in what follows. An immediate effect of increasing the source set, was a dramatic improvement in accuracy to 90.3%.

Measuring the percentage of "good" instances (i.e. instances with weight above a certain threshold) didn't reveal a strong correlation with the labels. In Figure 3 we show this result for a threshold of 0.8. As can be seen, labels 7-9 are slightly higher than the rest but there is no strong evidence of labels 5-9 being more useful than 0-4, as one might hope for.

That said, a more careful examination of low- and high-weighted instances, revealed that the usefulness of an instance, as determined by SOSELETO, is more tied to its appearance: namely, whether the digit is centered, at a similar size as MNIST, the amount of blur, and rotation. In Figure 4 we show a random sample of some "good" and "bad" (i.e. high and low weights, respectively). A close look reveals that "good" instances often tend to be complete, centered, axis aligned, and at a good size (wrt MNIST sizes). Especially interesting was that, among the "bad" instances, we found about $3 - 5\%$ wrongly labeled instances! In Figure 5 we display several especially problematic instances of the SVHN, all of which are labeled as "0" in the dataset. As can be seen, some examples are very clear errors. The highly weighted instances, on the other hand, had almost no such errors.

% instances above threshold

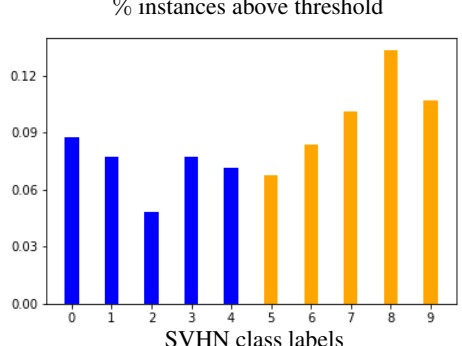

SVHN class labels

Figure 3: Percentage of good instances from SVHN per class. Classes 0-4 are colored blue and classes 5-9 are colored orange.

Randomly sampled "good" instances      Randomly sampled "bad" instances

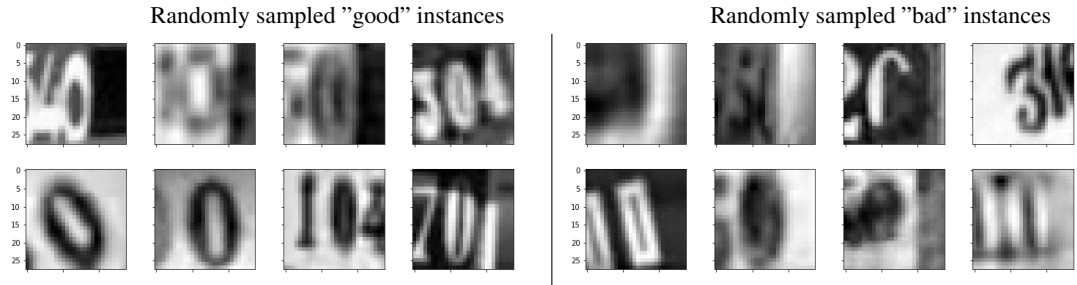

Figure 4: SVHN "good" (left) and "bad" (right) instances of class label 0.

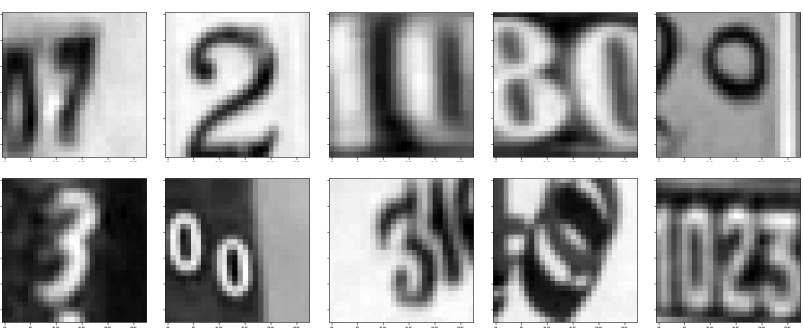

Figure 5: Hand-picked examples from the pool of "bad" instances with label 0.

