# OpenReview forum: "SOSELETO: A Unified Approach to Transfer Learning and Training with Noisy Labels"
_ICLR.cc/2019/Workshop/LLD — LLD 2019_

### Official Review · AnonReviewer2 · 2019-04-07
**Well motivated work with interesting idea**

**Rating:** 4
**Confidence:** 2

**Review:**

Pros:
- clear and concise writing, clear motivation for the transfer learning part
- sufficient detail presented for each experiment
- interesting idea to relate an approach for training with noisy labels to one for transfer learning, and show improvement on previous noisy label results

Cons:
- Figure 1 text is not readable (axis labels, legends, titles)

---

### Official Review · AnonReviewer1 · 2019-04-08
**Intuitive idea with solid execution. Promising experimental results**

**Rating:** 4
**Confidence:** 1

**Review:**

This paper presents a unified framework for transfer learning and learning with (large amounts of) noisily-labeled (source) data. The authors assume the source and target classifiers share the same “representation layers” and only differ by the “classification layers”.  They then jointly learn these layers as well as the source example weights. Specifically, the authors formulate the problem as a bilevel optimization problem and design the learning process such that the exterior level minimizes the target loss only through the source weights.  As a result, the target classifier can only control/adjust the features representation by adjusting the source example weights. Finally, the authors prove that under certain conditions this bilevel optimization procedure will converge and empirically show it helps learning.

Overall the paper is in good shape. I would like to see more (empirical) analysis on the convergence condition and understand how likely the condition at the end of page 9 will be satisfied. Also, some hyper-parameter sensitively analysis on lambda_{a} and lambda_{p} will be interesting. Finally, there are some minor mistakes (listed below) that need to be fixed:
1. in the equation (1), change min to argmin
2. in page 8 appendix A line 5, “A summer of SOSELETO algorithm appears in (Algorithm) 1”
3. in page 10 Appendix E line 2, Section “???”

---

### Decision · Program_Chairs · 2019-04-08
**Acceptance Decision**

Accept